# Synaptic Dysfunction by Mutations in *GRIN2B*: Influence of Triheteromeric NMDA Receptors on Gain-of-Function and Loss-of-Function Mutant Classification

**DOI:** 10.3390/brainsci12060789

**Published:** 2022-06-15

**Authors:** Marwa Elmasri, James S. Lotti, Wajeeha Aziz, Oliver G. Steele, Eirini Karachaliou, Kenji Sakimura, Kasper B. Hansen, Andrew C. Penn

**Affiliations:** 1Sussex Neuroscience, School of Life Sciences, University of Sussex, Brighton BN1 9QG, UK; marwa_elmasri@hotmail.com (M.E.); w.aziz@bsms.ac.uk (W.A.); o.steele@sussex.ac.uk (O.G.S.); ek310@sussex.ac.uk (E.K.); 2Center for Structural and Functional Neuroscience, Center for Biomolecular Structure and Dynamics, Division of Biological Sciences, University of Montana, Missoula, MT 59812, USA; james.lotti@umontana.edu (J.S.L.); kasper.hansen@mso.umt.edu (K.B.H.); 3Department of Cellular Neurobiology, Brain Research Institute, Niigata University, Niigata 951-8585, Japan; sakimura@bri.niigata-u.ac.jp

**Keywords:** ionotropic glutamate receptors, de novo mutations, central nervous system, synaptic transmission, electrophysiology

## Abstract

*GRIN2B* mutations are rare but often associated with patients having severe neurodevelopmental disorders with varying range of symptoms such as intellectual disability, developmental delay and epilepsy. Patient symptoms likely arise from mutations disturbing the role that the encoded NMDA receptor subunit, GluN2B, plays at neuronal connections in the developing nervous system. In this study, we investigated the cell-autonomous effects of putative gain- (GoF) and loss-of-function (LoF) missense *GRIN2B* mutations on excitatory synapses onto CA1 pyramidal neurons in organotypic hippocampal slices. In the absence of both native GluN2A and GluN2B subunits, functional incorporation into synaptic NMDA receptors was attenuated for GoF mutants, or almost eliminated for LoF GluN2B mutants. NMDA-receptor-mediated excitatory postsynaptic currents (NMDA-EPSCs) from synaptic GoF GluN1/2B receptors had prolonged decays consistent with their functional classification. Nonetheless, in the presence of native GluN2A, molecular replacement of native GluN2B with GoF and LoF GluN2B mutants all led to similar functional incorporation into synaptic receptors, more rapidly decaying NMDA-EPSCs and greater inhibition by TCN-201, a selective antagonist for GluN2A-containing NMDA receptors. Mechanistic insight was gained from experiments in HEK293T cells, which revealed that GluN2B GoF mutants slowed deactivation in diheteromeric GluN1/2B, but not triheteromeric GluN1/2A/2B receptors. We also show that a disease-associated missense mutation, which severely affects surface expression, causes opposing effects on NMDA-EPSC decay and charge transfer when introduced into GluN2A or GluN2B. Finally, we show that having a single null *Grin2b* allele has only a modest effect on NMDA-EPSC decay kinetics. Our results demonstrate that functional incorporation of GoF and LoF GluN2B mutants into synaptic receptors and the effects on EPSC decay times are highly dependent on the presence of triheteromeric GluN1/2A/2B NMDA receptors, thereby influencing the functional classification of NMDA receptor variants as GoF or LoF mutations. These findings highlight the complexity of interpreting effects of disease-causing NMDA receptor missense mutations in the context of neuronal function.

## 1. Introduction

*N*-methyl-D-aspartate receptors are ligand-gated cation channels that are highly abundant at excitatory synapses in the vertebrate central nervous system. NMDA receptors are tetrameric protein complexes that, by necessity, are heterooligomers containing a pair of obligatory GluN1 subunits together with two GluN2A-D or two GluN3A-B subunits [1,2]. NMDA receptor activation depends on the amino acid neurotransmitter glutamate and on ambient coagonist glycine or D-serine [3,4,5]. Ion permeation through the activated NMDA receptor channel pore is also dependent on membrane depolarization extruding a Mg^2+^, which otherwise chronically blocks the channel pore at resting membrane potential [6,7]. The open NMDA receptor channel pore is permeable to monovalent cations and the divalent cation, Ca^2*+*^ [8,9], which plays a pivotal role in the regulation of synaptic strength [10,11]. Genetic manipulations of NMDA receptor expression [12,13,14,15,16] or function [3,17,18,19] have been shown to have profound effects on behaviour [3,15,16,20,21] and survival [3,12,14,17,18,22].

The time course and extent of ion flux through NMDA receptor channels can be modified by receptor subunit composition [23,24,25,26], alternative splicing of NMDA receptor subunit gene transcripts [27,28,29], posttranslational modifications (e.g., phosphorylation) [30], and by interaction with endogenous allosteric ligands (e.g., extracellular Zn^2+^ and polyamines) [27,29,31]. NMDA receptors have also been shown to be intolerant to mutation [32], them being sensitive to single amino acid substitutions (missense mutations) that are linked to rare neurological and developmental disorders [2]. The genes encoding GluN2A and GluN2B, *GRIN2A* and *GRIN2B*, harbour mutations that are most frequently associated with disease from the NMDA glutamate receptor type, accounting for 43 and 35% of NMDA receptor mutations respectively [2]. Variation in symptoms between patients with *GRIN* mutations and the lack of drugs approved to treat them has prompted researchers to experimentally evaluate the effects of the mutations on NMDA receptor expression and function. These efforts have been encouraged by some cases where characterization of mutant receptor function and pharmacology has empowered clinicians to treat symptoms of *GRIN*-related disorders with off-label prescriptions [33,34,35,36,37,38,39].

Not all efforts to characterize mutations and to personalize treatment have resulted in improvement [34,39], and the developments tend to be relevant for some symptoms (e.g., seizures) more than others. Most evaluations of *GRIN* mutations are accomplished by expressing the mutated *GRIN* cDNA in heterologous expression systems. These systems can enable relatively simple and controlled investigations into the effect of mutations on receptor expression, biophysics and pharmacology. However, they do not recapitulate aspects of the native environment within which NMDA receptors normally operate. For example, the expression of (and co-assembly with) other NMDA receptor subunits, and a variety of endogenous modulatory ligands, all influence NMDA receptor function (e.g., pH, Zn^2+^ and co-agonists) [1,2].

Recently, we used molecular replacement experiments in cultured hippocampal slices to evaluate how *GRIN2A* mutations can affect basal synaptic function [40]. A surprising finding from this work was that completely different, i.e., gain-of-function (GoF) and loss-of-function (LoF), mutations led to surprisingly similar defects in synaptic transmission. An open question is whether the same would be observed for *GRIN2B* mutations. Many other labs have extensively characterized the effect of putative GoF and LoF mutants on the expression and function of GluN2B-containing NMDA receptors in heterologous expression systems [41] (Table 1). The GluN2B agonist binding domain mutants R696H and R540H form NMDA receptors with slow deactivation as well as overall greater charge transfer, indicative of the mutant channels staying open for longer following the removal of agonist and providing gain of function [42,43]. On the other hand another mutation in the same domain, C456Y, exhibited very poor surface expression and had a negative effect on channel activity [42,44]. Therefore, it is anticipated that R696H and R540H mutations would enhance the contribution of GluN2B to NMDA-EPSCs, while C456Y and other LoF mutations would diminish it (Table 1).

Here, we directly examined the consequences of GoF and LoF *GRIN2B* mutations on synaptic function in cultured mouse hippocampal slices by (molecular) replacement with the mutated human GluN2B subunits in CA1 neurons, in which expression of both native GluN2A and GluN2B was knocked out. We found that the putative GoF (R696H and R540H) but not LoF (C456Y) mutants show functional incorporation into synapses, and that their NMDA-EPSCs are more prolonged. However, in the continued presence of native GluN2A (i.e., with only native GluN2B knocked out), the NMDA-EPSCs for all mutants had similar defects, notably an acceleration of NMDA-EPSC decay. By examining the block of NMDA-EPSCs by GluN2A antagonist TCN-201, and by measuring the deactivation of GoF GluN2B mutations in heterologous expression systems, we found that the NMDA-EPSC phenotype results in part from an abolishment of the GluN2B mutation effect when expressed as triheteromers together with GluN2A (i.e., GluN1/2A/2B). We also provide results highlighting the importance of the identity of the subunit that is mutated (rather than the mutation itself) for its effect on NMDA-EPSCs. Similar to our findings for GluN2A [40], we also demonstrate partial haploinsufficiency for neurons that are heterozygous for LoF alleles of *Grin2b*.

## 2. Materials and Methods

### 2.1. Molecular Biology

Mutagenesis was performed on the cDNA of human *GRIN2B* in the pCI-neo vector (pCMV GluN2B). GluN2B mutations with respect to the amino acid numbering in NP_000825.1 were: C436R, C456Y, R696H, R540H. Oligonucleotide primers (Integrated DNA Technologies, BVBA, London, UK) were designed to incorporate single-base substitutions using the GeneArt^®^ Site-Directed Mutagenesis Kit (Fisher Scientific UK Ltd., Loughborough, UK) (Appendix A). The presence of each mutation was confirmed by DNA sequencing (Eurofins Genomics, Wolverhampton, UK) and the whole coding sequence was also examined to ensure that no other mutations were erroneously introduced during polymerase chain reaction (PCR). Plasmid DNA was amplified and purified using EZNA endo-free plasmid maxi kit (VWR International, Lutterworth, UK).

Rat cDNAs for GluN1-1a (GenBank accession number U08261; hereafter GluN1) GluN2A (GenBank accession number D13211), and GluN2B (GenBank accession number U11419) were used for NMDA receptor expression in HEK293T cells. As previously described [28], the DNA construct for expression of rat GluN1 contained the open-reading-frame (ORF) for enhanced green fluorescent protein (EGFP) inserted between the CMV promoter in pCI-neo and the ORF for GluN1 (i.e., EGFP and GluN1 were not expressed as a fusion protein). The placement of 0–3 AUGs, which initiate short upstream ORFs encoding 42–58 amino acids, between EGFP and GluN1 in this DNA construct results in a gradual reduction of GluN1 expression, while maintaining high expression of EGFP. DNA constructs for expression of GluN2 subunits in diheteromeric GluN1/2A and GluN1/2B receptors as well as triheteromeric GluN1/2A/2B receptor have previously been described in detail [24,47].

Amino acids are numbered according to the full-length protein, including the signal peptide.

### 2.2. Animals

C57Bl/6J mice and Sprague Dawley rats were purchased from Charles River Laboratories. Heterozygous *Grin2a*-flox and *Grin2b*-flox mice were gifted by Kenji Sakamura (Niigata University Japan) [48,49] and, through mating of respective lines, maintained as homozygous lines in-house. Double-flox mice were bred through crossing of these lines. All mice were conventionally housed and were in a 12-12 light–dark cycle throughout. Genotype was confirmed prior to experimental use via PCR, as described previously [40]. In accordance with the Animals Scientific Procedures Act 1986 (amended in 2012), neonatal pups were culled for use in experiments by trained and approved personnel via cervical dislocation, an appropriate Schedule 1 procedure. All experiments were reviewed by the Animal Welfare Ethical Review Body at the University of Sussex and performed within a Home Office project license (PPL# P6CF775B9).

### 2.3. Slice Culture and Transfection

Organotypic hippocampal brain slices were prepared from mice between postnatal days 6–8. Following cervical dislocation, hippocampi were dissected in ice-cold sterile filtered dissection media containing (in mM): CaCl_2_ (0.5), KCl (2.5), KH_2_PO_4_ (0.66), MgCl_2_ (2), MgSO_4_ (0.28), NaCl (50), Na_2_HPO_4_ (0.85), glucose (25), NaHCO_3_ (2.7), sucrose (175) and HEPES (2) (pH 7.3, 330 mOsm). Transverse slices (350 µm) were cut using a McIlwain tissue chopper (Abbotsbury Engineering Ltd., St. Ives, UK), transferred to pieces of membrane (FHLC01300, Millipore, UK) and placed in Millicell cell culture inserts (PICM03050, Millipore, UK) in 6-well plates. Slices were then maintained at 34 °C with 5% CO_2_ and 95% humidity in a Minimum Essential Media (MEM)-based culture medium (1 mL per well) supplemented with 15% heat-inactivated horse serum (26050-088, Gibco), 1x B27 supplement (17504044) as well as (in mM): HEPES (25), L-glutamine (3), CaCl_2_ (1), MgSO_4_ (1), ascorbic acid (0.25) and glucose (27.75). The culture medium was changed twice per week, with the inclusion of 10 µg/mL gentamicin after DIV7.

CA1 pyramidal neurons in organotypic slice cultures were transfected by single cell electroporation between 6–8 days in vitro (DIV) [40,50]. Neurons were electroporated with plasmids expressing pCMV Cre-GFP (3 ng/μL 0.58 nM) with or without WT or mutant (R540H, R696H, C456Y, C461F and C436R) pCMV *GRIN2B* cDNA (10 ng/μL, 1.59 nM). Plasmid DNA was precipitated upon the addition of 0.5 volume of PEG solution (30% PEG 8000 (*w*/*v*) in 30 mM MgCl_2_ before centrifugation at 10,000 relative centrifugal force (rcf) for 15 min. The pellet was then washed with 70% ethanol, air-dried and then re-dissolved in TE buffer (10 mM Tris-HCl, 1 mM EDTA). Plasmids were diluted to 0.53 nM (3 ng/μL) in intracellular solution containing (in mM): CH_3_SO_3_H (135), KOH (135), NaCl (4), MgCl_2_ (2), HEPES (10), NaATP (2), NaGTP (0.3), spermine dihydrate (0.15), EGTA (0.06) and CaCl_2_ (0.01) (pH 7.25, 285 mOsm). Debris was removed from DNA-containing intracellular solution via centrifugation at >10,000 rcf before being loaded into 8–10 MΩ patch pipettes pulled from thick-walled borosilicate glass capillaries (GB150F-8P: Science Products GmbH, Hofheim, Germany). Slices were then submerged in room temperature extracellular recording solution containing (in mM): NaCl (140), KCl (3), MgCl_2_ (1), CaCl_2_ (2), glucose (10), Na-pyruvate (1), NaHCO_3_ (2), HEPES (6) and Na-HEPES (4) (pH 7.35, 300 mOsm). Filled patch-pipettes were then positioned in CA1 using an upright microscope (Olympus BX51: Scientifica Ltd., Uckfield, UK) and manual manipulator (PatchStar: Scientifica Ltd., Uckfield, UK). Positive pressure (~20 mbar) was applied during approach to pyramidal cells and upon dimple formation pressure was released to form a loose-patch seal. A 12 V stimulus train (100–200 Hz for 0.25–0.5 s; pulse width 0.25–0.5 ms) was then applied via a stimulus isolator (Isoflex: A.M.P.I., Jerusalem, Israel) triggered by ACQ4 software (v0.9.3) [51] through a USB-X Series Multifunctional DAQ interface (NI USB-6341: National Instruments Corp Ltd., Newbury, UK). With the same patch pipette, transfection of ten CA1 pyramidal cells was attempted in each slice, with successful expression (judged by the presence of fluorescence) observed half the time.

### 2.4. Slice Electrophysiology

Evoked current responses were measured using whole-cell patch-clamp. Hippocampal slices, featuring a cut between CA3 and CA1 to reduce recurrent excitation, were transferred to the sample chamber on an upright microscope (SliceScope Pro 2000: Scientifica Ltd., Uckfield, UK) and imaged using infrared light (780 nm) through an oblique condenser and 40 × 0.8 NA objective. Slices were submerged in 32 °C heated oxygenated (95% O_2_, 5% CO_2_) artificial cerebrospinal fluid (aCSF) containing (in mM): NaCl (125), KCl (2.5), NaHCO_3_ (25), glucose (10), Na-pyruvate (1), MgCl_2_ (1), CaCl_2_ (2) and 2-chloroadenosine (0.005), which was recirculated at rate of 2 mL/min. Schaffer collaterals were stimulated with a bipolar tungsten stimulating electrode positioned in stratum radiatum, and CA1 pyramidal neurons were recorded in whole-cell configuration with 2–4 MΩ patch pipettes filled with intracellular solution containing (in mM): CH_3_SO_3_H (120), CsOH (120), CsCl (20), MgCl_2_ (2.5), HEPES (10), NaATP (4), NaGTP (0.4), phosphocreatine disodium salt (5) and EGTA (0.2) (pH 7.25, mOsm 295). Two neurons, one transfected (fluorescent) and a neighbouring non-transfected neuron, were patched simultaneously. Constant voltage stimulus (50 μs duration, 10 s interval) strength and polarity were adjusted (IsoFlex: A.M.P.I., Jerusalem, Israel) to obtain reliable NMDA-EPSCs in non-transfected neurons that were <250 pA (typically 10 V). Reponses were then acquired with a MultiClamp 700B (Molecular Devices, Wokingham, UK), low-pass filtered (4 kHz, 4-pole Bessel), and digitised using a USB-X Series Multifunctional DAQ Interface (NI USB-6363: National Instruments Corp Ltd., Newbury, UK) controlled by ACQ4 software (v0.9.3) [51]. Series resistance was left uncompensated during acquisition but compensated offline (see below) and discarded if resistance changed by >8–10 MΩ. All reported holding potentials account for the liquid–liquid junction potential, which we calculated to be +10 mV for our solutions (PowerPatch Tools Igor Pro Tool collection).

Recording protocols for pharmacologically isolated NMDA-EPSCs are exactly as described previously [40]. AMPA-EPSCs were measured at a holding potential of −100 mV. NMDA-EPSCs were calculated from the synaptic conductance measured at +20 mV by subtraction of the AMPA-EPSC conductance as described [40]. For TCN-201 experiments a 50 μs square pulse stimulus (20–40 V) was repeated (20 repetitions; 10 s cycle time) on the Schaffer collaterals and NMDAR-EPSCs were recorded postsynaptically from transfected CA1 neurons. This recording was followed by addition of 10 μM TCN-201 (Bio-techne Ltd., Abingdon, UK) to the aCSF. One recording was taken every 30 s during this period, and finally another recording of 20 traces was taken at 17.5 min after drug application.

We performed offline corrections for differences or changes in series resistance [52]. The stimulus artefact was interpolated, and the series resistance (*Rs*) and cell membrane capacitance were used to perform offline series resistance compensation. For consistency, final, uncompensated resistance in each recording (*Rs_final_*) was set to a constant 3 MOhm, where the % of compensation applied to each recording was calculated using Equation (1):(1)$$Rs_{comp}\left(\%\right)=100\times \max\left(\;1-\frac{Rs_{final}}{Rs}\;,\;0\;\right)$$

Series-resistance-compensated recording traces were analysed using custom scripts in Matlab (Mathworks Ltd., Cambridge, UK) available at https://github.com/acp29/Elmasri_GRIN2B (accessed on 5 May 2022). EPSC current decays were fit by a two-component exponential decay with offset using the Chebyshev algorithm [53]. The weighted exponential time constant (*τ_w_*) was calculated as the sum of component time constants multiplied by their fractional amplitude according to Equation (2)
(2)$$\tau_{w}=\left(\tau_{fast}\times I_{fast}+\tau_{slow}\times I_{slow}\right)/\left(I_{fast}+I_{slow}\right)$$
where *τ_fast_* and *τ_slow_* are the deactivation time constants and *I_fast_* and *I_slow_* are the current amplitudes for the fast and slow components, respectively. Where the two-component fits failed, a single exponential was found to fit well.

### 2.5. HEK293T Transfection and Whole-Cell Patch-Clamp Electrophysiology

Approximately 48 h before experiments, HEK293T cells were plated onto poly-D-lysine-coated (0.1 mg/mL) glass coverslips and cultured in Dulbecco’s modified Eagle’s medium with GlutaMax-I and sodium pyruvate supplemented with 10% dialyzed foetal bovine serum, 10 U/mL penicillin and 10 µg/mL streptomycin (GIBCO; Thermo Fisher Scientific). Transfection was performed the following day using the calcium phosphate precipitation method as described previously [54], with DNA constructs encoding GluN1 (together with EGFP in the pCI-neo vector) and GluN2 subunits at a ratio of 1:1. For triheteromeric NMDA receptors, DNA constructs encoding GluN1 and the two GluN2 subunits were transfected at a 1:1:1 ratio. NMDA-receptor-mediated cytotoxicity was prevented by the addition of antagonists 7-chlorokynurenic acid (200 µM) and D,L-2-amino-5-phosphonovalerate (200 µM) to the culture medium.

Whole-cell voltage-clamp recordings were conducted approximately 24 h following transfection at room temperature (21–23 °C) using an Axopatch 200B amplifier (Molecular Devices) and a holding potential of –60 mV. The signal was filtered using an 8 kHz 8-pole low-pass filter (−3 dB Bessel; Frequency Devices), digitized using Digidata 1322A (Molecular Devices) at 20 kHz and recorded using pCLAMP software (Molecular Devices). Recording electrodes with open-tip resistance of 2–4 MΩ were pulled from thin-wall glass micropipettes (TW150F-4: World Precision Instruments LLC, Sarasota, FL, USA) using a horizontal puller (P-1000: Sutter Instrument, Novato, CA, USA). Recording electrodes were filled with internal solution containing (in mM): D-gluconic acid (110), CsOH (110), CsCl (30), HEPES (5), NaCl (4), CaCl_2_ (0.5), MgCl_2_ (2), 1,2-bis(2-aminophenoxy)ethane-N,N,N′,N′-tetraacetic acid (5), NaATP (2) and NaGTP (0.3) (pH 7.35 with CsOH). The extracellular recording solution consisted of 150 mM NaCl, 10 mM HEPES, 3 mM KCl, 0.5 mM CaCl_2_, and 0.01 mM EDTA (pH 7.4 with NaOH). Rapid solution exchange was achieved on lifted cells using a theta-glass pipette controlled by a piezoelectric translator (MXPZT-300: Siskiyou Corporation, Grants Pass, OR, USA). The open-tip solution exchange times (10–90%) were 0.6–0.8 ms. Series resistance was not compensated, but only cells with current responses of less than 1000 pA and recordings with a series resistance of less than 10 MΩ were used for data analyses.

Macroscopic response time courses of NMDA receptor responses expressed in HEK293T cells were analysed using Axograph X (Axograph Scientific, Sydney, Australia). The deactivation time courses of current responses were fitted using Equation (3)
(3)$$I_{total}=I_{fast}\times e^{\left(-time/\tau_{fast}\right)}+I_{slow}\times e^{\left(-time/\tau_{slow}\right)}$$

*τ_fast_* and *τ_slow_* are the deactivation time constants and *I_fast_* and *I_slow_* are the current amplitudes for the fast and slow components, respectively. The weighted deactivation time constants (*τ_w_*) were calculated using Equation (2).

### 2.6. Statistics

Data were collected until the number of cell pairs reached a total of between about 12–37 per mutation (or genotype), which were obtained from 9–19 slices and 2–11 animals. Animals and/or slices were randomly assigned to GluN2B mutants in rescue experiments and the experimenter was blind to the mutation. Outlying cell recordings were excluded from subsequent statistical analysis by robust principal component analysis (ROBPCA) using the LIBRA Matlab library [55,56] in Matlab 9.2 (R2017a: Mathworks, Ltd., Cambridge, UK), as described in [40]. Circa 2 recording pairs (range from 0 to 4) were excluded per group of data (e.g., per mutation). A summary of the sample sizes after multivariate outlier detection is presented in Appendix A.

Statistical analysis was carried out in R (v 4.1.0) and RStudio (v 1.4) as described in [40]. In brief, each response datum from patch-clamp recordings (in Figure 1, Figure 2, Figure 3, Figure 5, Figure 6 and Appendix A) was collected along with its assignment to (levels of) the following factors: mutation (e.g., WT, R696H, C456Y, etc.) or genotype (WT, heterozygote, homozygote), transfection (unt. or tr., for untransfected or transfected, respectively) or drug (before or after), and cell or (cell) pair, slice, and animal. Factors were appropriately assigned as fixed effects, or (nuisance) random effects. Response data from each experimental dataset were log_e_-transformed and then fit with univariate linear mixed models (LMMs) using the lmer function from the R package lme4. Model fitting was set to optimise the restricted maximum likelihood (REML) criterion. The lmer formula for each random intercept model is reported in Appendix A and follows the described convention for mixed models [57]. Where hypothesis testing was based on a priori clustering of the data, orthogonal contrasts (Appendix A) were set up and applied before model fitting. Where hypothesis testing was based on testing the significance of trends across ordered (categorical) factors (e.g., in Figure 6), polynomial contrasts were set up and applied before model fitting. Model assumptions were evaluated by graphing plots of the residuals as previously described [40].

The LMM results for the fixed effects were summarised as ANOVA tables using Type III Wald F-tests and with degrees of freedom calculated by the Kenward–Roger method. To summarise the random effects structure, the variance of each group level was used to calculate its respective intraclass correlation coefficient (ICC). Parameter estimates and effect sizes are reported as response ratios (expressed as %) with 95% confidence intervals in the main text and figures. Effect sizes were standardised by conversion to correlation coefficients and reported in the figure legends. Details of LMM results and diagnostics are provided in the Appendix A. All summaries of statistics annotated in graphs used the following convention: ns = not significant (at α = 0.05), * = *p* < 0.05, ** = *p* < 0.01, *** = *p* < 0.001. All reported *p*-values are from two-tailed tests. We also calculated Bayes Factors (BF_10_) to evaluate the strength of evidence for the null hypothesis in ANOVA tests of the fixed effects. Code and data relating to the analysis are provided at https://github.com/acp29/Elmasri_GRIN2B (accessed on 5 May 2022). Further details relating to the statistical methods can also be found in [40].

## 3. Results

### 3.1. GoF GRIN2B Mutants Can Traffic to Synapses of Grin2a^−/−^b^−/−^ Neurons and Produce NMDA-EPSCs with Prolonged Time Course

We first sought to establish whether putative GoF and LoF GluN2B missense mutants can be incorporated into synapses and contribute to synaptic currents. We introduced R540H, R696H and C456Y mutations listed in Table 1 into the human GRIN2B cDNA and coexpressed them with Cre-GFP in CA1 neurons of organotypic slices prepared from *Grin2a^fl/fl^b^fl/fl^* mice (Figure 1a). NMDA-EPSCs evoked by stimulation of the Schaffer Collateral axons, which form synaptic contacts onto CA1 neurons, are largely contributed by GluN2A- and GluN2B-containing NMDA receptors, as evident from the drop in NMDA-EPSC amplitude in Cre-GFP positive neurons to 12% of the NMDA-EPSC amplitude in untransfected neurons (95% CI [9–16%]) (Figure 1b,c). Coexpressing WT GluN2B with Cre-GFP rescued NMDA-EPSC amplitude to 51% (95% CI [39–68%]) compared to in untransfected neurons (100%). The incomplete rescue by GluN2B alone is expected since both GluN2A and GluN2B make similarly large contributions to NMDA-EPSCs at Schaffer collateral synapses [40,49]. The increase in size of the NMDA-EPSC by rescuing with GluN2B WT corresponded to the current peak increasing to 418% (95% CI [283–618%]) relative to *Grin2a*^−/−^*b*^−/−^ (GluN2A/B double knockout, DKO) neurons (Figure 1: None, for exogenous no GluN2A). We could also resolve significant increases in NMDA-EPSC peak amplitude between DKO neurons (100%) and DKO neurons rescued with GluN2B R540H and R696H, by 241% (95% CI [154–376%]) and 189% (95% CI [126–285%]), respectively. Relative to untransfected neurons, this corresponded to GluN2B R540H and R696H restoring NMDA-EPSC peak amplitudes to 30% (95% CI [21–42%]) and 23% (17–32%). The smaller peak currents of the R540H and R696H mutants compared to WT GluN2B are consistent with reports of slightly lower surface expression and current density when expressed in HEK cells (Table 1) [42]. Equally in agreement with published findings using heterologous expression systems, the NMDA-EPSC amplitude for the LoF mutant C456Y was only 120% (95% CI [83–173%]) that of DKO neurons (100%, Figure 1).

We next compared the amount of charge passed during the NMDA-EPSCs. Charge transfer of R540H and R696H was not significantly different from WT, with 77% (95% CI [45–130%]) and 70% (95% CI [43–114%]) of the size in WT neurons (100%) respectively. In contrast, charge transfer in DKO neurons rescued with GluN2B C456Y was only 19% the size of NMDA-EPSC charge transfer in WT neurons (Figure 1d and Appendix A). This suggests that the R540H and R696H mutants are not obviously GoF mutations, however, upon peak scaling the NMDA-EPSCs, we noticed that the decay time constant was significantly different among WT, R540H and R696H (*F* (1,90) = 19.20, *p* = 0.03), with R540H being 151% (95% CI [106–216%]) and R696H 146% (95% CI [105–204%]) that of WT (Figure 1e and Appendix A)

Together, these data indicate that while GoF features of NMDA-EPSCs are evident for R540H and R696H mutants, their similar charge transfer to WT suggests that they could have a complex synaptic phenotype.

### 3.2. LoF and GoF GluN2B Mutant Rescue in Grin2b^−/−^ Neurons Can Produce Similar Effects on NMDA-EPSCs

The putative GoF and LoF GluN2B missense mutants varied in their ability to contribute functional receptors at synapses (Figure 1). To understand how the mutations are likely to affect synaptic transmission with native GluN2A expression intact, we next assessed the effects of rescuing synaptic currents in *Grin2b* knockout neurons by introducing WT or mutant GluN2B subunits. In these experiments, we included another LoF mutant, C461F, which shows reasonable surface expression in neurons but has more than a 100-fold drop in glutamate potency, for comparison [42]. The peak amplitude and decay time of NMDAR EPSCs in GluN2B KO neurons was effectively restored by 10 ng/µL of a plasmid expressing WT human GluN2B from a CMV promoter (Appendix A). Using the same vector and plasmid concentration, we found that NMDA-EPSCs by mutant GluN2B were similar in amplitude to WT, with no effect detected among the peak amplitudes across the groups (Figure 2a, Appendix A, BF_10_ = 0.052). Nonetheless, relative to WT, estimates of the peak NMDA-EPSCs amplitudes appeared slightly smaller for LoF mutants (91% for C456Y and 86% for C461F, 95% CIs [64–128%] and [60–124%], respectively), than for GoF mutants (97% for R540H and 94% for R696H, 95% CIs [71–133%] and [66–133%], respectively). Similarly, we found no significant effect of the GluN2B mutations on either AMPA-EPSC peak amplitudes (Appendix A; Appendix A: *F* (4,90) = 1.17, *p* = 0.33, BF_10_ = 0.197) or AMPA/NMDA ratios (*F* (4,90) = 1.17, *p* = 0.38, BF_10_ = 0.167). These results suggest that, in the presence of native GluN2A, the functional incorporation of mutant GluN2B subunits into synaptic receptors is comparable to WT GluN2B. This finding is in stark contrast to the reduced functional incorporation of GoF and LoF mutant GluN2B subunits in neurons lacking both native GluN2A and native GluN2B (Figure 1b,c).

While examining the NMDA-EPSCs, we noticed that the decay kinetics appeared to be faster in the mutants (Figure 2bi). After comparing the time constant of the NMDA-EPSCs, we found supporting evidence for an effect of GluN2B mutation on NMDA-EPSC decay (Appendix A, *F* (4,90) = 8.85, *p* < 0.001, BF_10_ = 1.52 × 10^5^). Overall, the decay time constants of the mutants were significantly different from WT (Figure 2b and Appendix A, *F* (1,90) = 19.20, *p* < 0.001) as well as between GoF and LoF mutation types (Figure 2b and Appendix A, *F* (1,90) = 17.17, *p* < 0.001). Surprisingly, the decay time constants were slightly smaller for GoF mutants relative to WT (100%), with 73% for R540H and 67% for R696H, 95% Cis [52–102%] and [46–97%], respectively. The effect was larger for LoF mutants and in the same direction, with the decay time constants for LoF mutants relative to WT (100%) being 44% for C456Y and 37% for C461F, 95% Cis [30–63%] and [25–54%], respectively.

Having detected changes in the decay of NMDA-EPSCs, we next assessed whether there would be any effect of *GRIN2B* mutations on NMDA-EPSC charge transfer by taking the integral of the current traces (Figure 2ci). Significant differences could be resolved in orthogonal contrasts comparing all mutants with WT (Figure 2c and Appendix A, *F* (4,90) = 3.63, *p* = 0.009), and between the GoF and LoF mutation types (Figure 2c and Appendix A, *F* (4,90) = 4.81, *p* = 0.031). Similar to the decay time constants, compared to WT (100%), NMDA-EPSC charge transfer was lower for LoF mutations C456Y (56%) and C461F (44%), 95% CIs [34–91%] and [27–74%], respectively, than it was for GoF mutations R540H (83%) and R696H (90%), 95% CIs [53–130%] and [55–148%], respectively (Figure 2c).

Overall, the effects of LoF and GoF GluN2B mutations on NMDA-EPSC decay kinetics were significant and in the same direction, generally resulting in smaller charge transfer of the NMDA-EPSC.

**Figure 2 brainsci-12-00789-f002:**
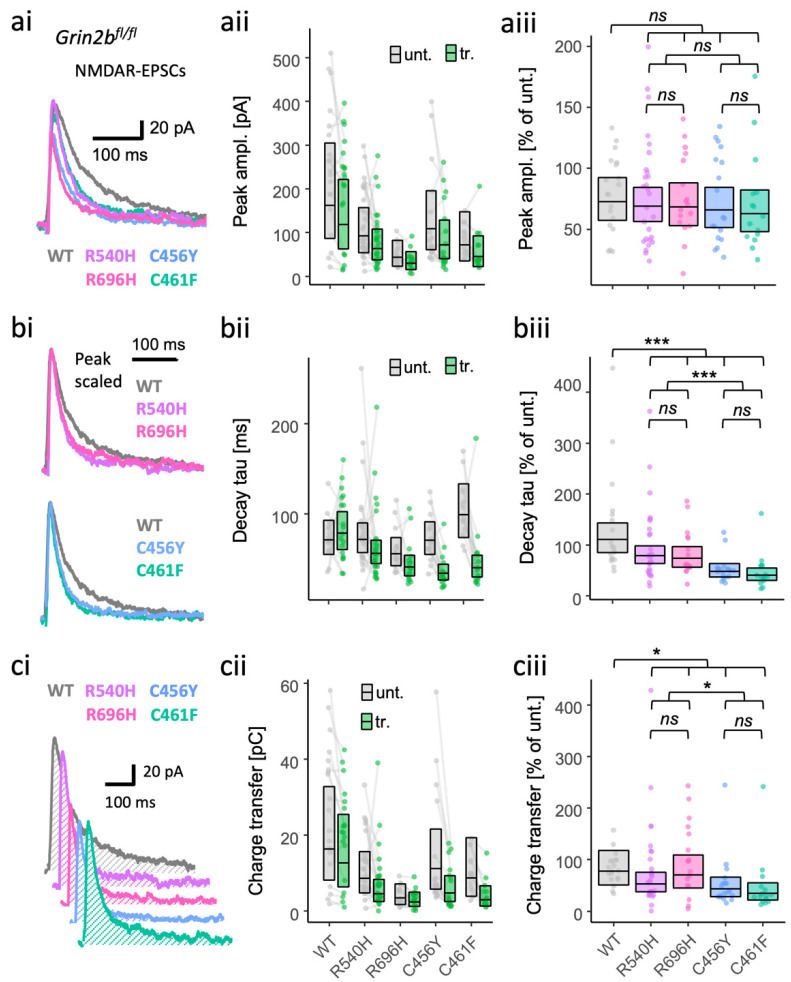
Gain- and loss-of-function GluN2B mutants are both associated with more rapidly decaying NMDA-EPSCs in GluN2B knockout neurons. NMDA-EPSC_+20 mV_ peak amplitudes (**a**), decay time constants (**b**) or charge transfer (**c**) in *Grin2b^fl/fl^* (untransfected) neurons and *Grin2b*^−/−^ neurons rescued with human GluN2B WT, GoF (R540H and R696H) or LoF (C456Y or C461F) mutants (transfected). (**a**–**c**) (i) Representative NMDA-EPSCs (average of 10 sweeps) from transfected CA1 neurons; (ii) data points of measurements made in individual neurons, matched data points, for simultaneously recorded untransfected and transfected neurons, are connected by a line; (iii) response ratios (transfected/untransfected) are expressed as a percentage and plotted for each pair of transfected–untransfected neurons. Crossbars in (ii) and (iii) show the estimated marginal means with 95% confidence intervals backtransformed from the linear mixed models (Appendix A). Hypothesis tests are orthogonal contrasts based on a priori classification of the mutations (see Appendix A). Standardised effect sizes (*r*) for comparisons of each mutant with WT for response ratios of: (**aiii**) peak amplitudes were −0.02, −0.04, −0.06, and −0.08; (**biii**) decay taus were −0.19, −0.22, −0.42, and −0.47; and (**ciii**) charge transfer was −0.09, −0.04, −0.24, and −0.31, for mutants R540H, R696H, C456Y, and C461F, respectively (N = 95). *ns* = not significant (at α = 0.05), * = *p* < 0.05, *** = *p* < 0.001.

**Figure 3 brainsci-12-00789-f003:**
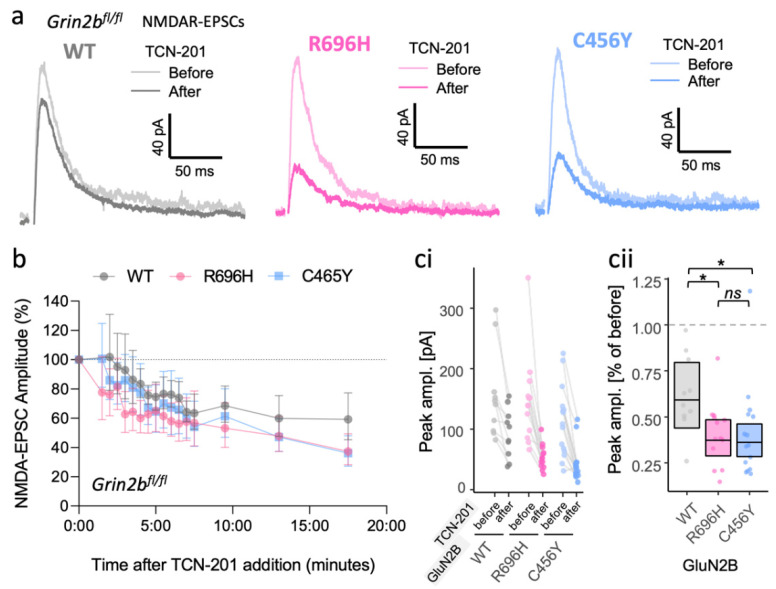
Gain- and loss-of-function GluN2B mutants both increase inhibition of NMDA-EPSCs in GluN2B knockout neurons by the GluN2A-selective antagonist TCN-201. (**a**) Representative pharmacologically isolated NMDA-EPSCs (average of 20 sweeps) from transfected CA1 neurons before and 17.5 min after application of 10 μM TCN-201. (**b**) NMDA-EPSC_+20 mV_ peak amplitudes expressed as a % of the initial EPSC size (before application of 10 μM TCN-201) and summarised for all time points as the geometric mean and 95% confidence intervals. (**ci**) NMDA-EPSC peak amplitudes before and 17.5 min after the addition of 10 µM TCN-201; each point represents the peak of the ensemble mean of 20 NMDA-EPSCs, each pair of points connected by a line corresponds to measurements made from individual neurons before and after addition of TCN-201; (**cii**) response ratios (transfected/untransfected) calculated from (**ci**) are expressed as a percentage and plotted for each pair of before–after recordings. Crossbars show the estimated marginal means with 95% confidence intervals backtransformed from the linear mixed models (Appendix A). Standardised effect sizes (*r*) for each mutant compared to WT were −0.37 and −0.40 for mutants R696H and C456Y, respectively (N = 38). *ns* = not significant (at α = 0.05), * = *p* < 0.05.

### 3.3. Inadequate Functional Synaptic Incorporation of GluN1/2B and Co-Assembly with GluN2A in GluN1/2A/2B Triheteromers Result in Similar Phenotypes for GluN2B GoF and LoF Mutants

We were surprised to find that *Grin2b*^−/−^ neurons rescued with the GoF mutants R540H and R696H produced NMDA-EPSCs that decayed more rapidly than those rescued with WT GluN2B. This contrasts with the slower-than-WT decays of NMDA-EPSCs in *Grin2a*^−/−^*b*^−/−^ neurons rescued with the same mutants. One possibility is that the relatively lower functional incorporation of GoF mutants results in more of the NMDA-EPSC being mediated by GluN1/2A or GluN1/2A/2B receptors, which have faster kinetics compared to GluN1/2B receptors [24]. In addition, it has been demonstrated that features of the GluN2A subunit can dominate functional properties of triheteromeric GluN1/2A/2B receptors [24,58,59], which may mitigate effects of mutant GluN2B subunits in the presence of GluN2A. Functional incorporation of mutant GluN2B into GluN1/2A/2B triheteromeric receptors was not possible in the experiments in Figure 1, since these neurons lack GluN2A expression. To assess the contribution of GluN2A-containing receptors to NMDA-EPSCs, we evaluated the extent to which a GluN2A selective antagonist, TCN-201, inhibited NMDA-EPSCs in *Grin2b*^−/−^ neurons rescued with WT GluN2B or with mutants R696H and C456Y. In nominal glycine, TCN-201 at 10 μM is an effective antagonist of GluN2A-containing NMDA receptors [24,60]. We found that 7.5 min after commencing wash-in of TCN-201, the NMDA-EPSCs peak amplitude for GluN2B WT was inhibited to 59% (95% CI [44–80%]) and remained stable for 10 min (Figure 3). Consistent with a greater contribution of GluN2A-containing NMDA receptors for LoF mutants with poor functional incorporation (Figure 3), residual NMDA-EPSC current for the C456Y mutant after TCN-201 application was lower at 36% (95% CI [28–46%]) (Figure 3). Similarly, the residual NMDA-EPSC current for GoF R696H mutant in *Grin2b*^−/−^ neurons was 37% (95% CI [29–48%]), which was comparable in size to the LoF mutant C456Y (Figure 3). The mutants differed significantly from WT (Figure 3c, Appendix A: *F* (1,35) = 7.81, *p* = 0.008) but not from each other with respect to NMDA-EPSC peak amplitude (Figure 3c, Appendix A: *F* (1,35) = 0.03, *p* = 0.86). The increased TCN-201 inhibition for GluN2B mutants compared to WT is consistent with a mechanism in which both GluN2B mutations reduce the contribution of GluN1/2B to the NMDA-EPSC. This mechanism is supported by results in Figure 1 that demonstrate reduced functional synapse incorporation for mutant GluN2B in diheteromeric GluN/2B receptors and the results in Figure 2 that show normal incorporation in the presence of native GluN2A subunits, potentially as triheteromeric GluN1/2A/2B.

The incomplete block of NMDA-EPSCs by TCN-201 in neurons rescued with the mutants indeed suggests that a large contribution of triheteromeric GluN1/2A/2B receptors exist, for which TCN-201 is a less potent and efficacious antagonist compared to at GluN1/2A diheteromers [24]. Since trihetereomeric GluN1/2A/2B receptors have rapid kinetics similar to GluN1/2A diheteromers, the contribution of GluN1/2B diheteromers may be revealed by observing an increase in the decay time of the NMDA-EPSC following TCN-201 application. Although we observed a trend for a small (~5–15%) increase in the decay time constant following TCN-201 treatment, this increase was not significant (Appendix A; *F* (1,35) = 3.03, *p* = 0.09, BF_10_ = 0.874), suggesting that (TCN-201 insensitive) GluN1/2B diheteromers make, if anything, only a minor contribution to the NMDA receptor population at Schaffer collateral synapses. In addition, while the increase in the decay time constant for mutants appeared smaller than for WT (Appendix A), we could not resolve a significant drug x mutant interaction (*F* (2,35) = 0.21, *p* = 0.81, BF_10_ = 0.187). Therefore, greater inhibition of the NMDA-EPSC peak amplitude in neurons rescued with the GluN2B mutants may also reflect a partial shift in the proportion of synaptic NMDA receptors from GluN1/2A/2B triheteromers to the more TCN-201-sensitive GluN1/2A diheteromers.

To determine functional effects of the GoF GluN2B mutants (R540H and R696H) in triheteromeric GluN1/2A/2B receptors, we evaluated the time course of current responses from NMDA receptor subtypes expressed in HEK293T cells using fast-application whole-cell patch-clamp recordings (Figure 4). The expression of triheteromeric GluN1/2A/2B receptors was facilitated by engineered C-terminal domains that only allow surface expression of receptors containing both GluN2A and GluN2B subunits due to mutual unmasking of the introduced endoplasmic reticulum retention signals [24,47]. The decay of responses activated by brief 5 ms exposures to 1 mM glutamate in the continuous presence of 100 µM glycine was similar for WT diheteromeric GluN1/2A and triheteromeric GluN1/2A/2B receptors, and markedly slower for diheteromeric GluN1/2B receptors (Figure 4a). The GoF mutations significantly slowed decay time in GluN1/2B^R540H^ and GluN1/2B^R696H^ compared to WT GluN1/2B receptors (Figure 4b and Appendix A), consistent with results shown in Figure 1. By contrast, the GoF mutations did not significantly change the decay time in triheteromeric GluN1/2A/2B^R540H^ and GluN1/2A/2B^R696H^ compared to WT GluN1/1A/2B receptors (Figure 4b and Appendix A). TCN-201 (5 µM) produced 95% inhibition of responses from WT GluN1/2A activated by brief 5 ms exposures to 1 mM glutamate in the continuous presence of 3 µM glycine, whereas TCN-201 only reduced responses from WT GluN1/2B by 4% (Figure 4c, Appendix A). surprisingly, TCN-201 inhibited responses from GluN1/2B^R540H^ by 19%, whereas inhibition of GluN1/2B^R696H^ was only 4% (Figure 4c, Appendix A). TCN-201 inhibited WT triheteromeric GluN1/2A/2B by 71% and GluN1/2A/2B^R540H^ and GluN1/2A/2B^R696H^ were inhibited by 83% and 74%, respectively (Figure 4c, Appendix A). Importantly, the decay time of current responses for WT and mutant triheteromeric GluN1/2A/2B receptors did not significantly change following TCN-201 (*F* (1,26) = 0.032, *p* = 0.86) (Appendix A), indicating that a population of diheteromeric GluN1/2B receptors are not enriched in these cells by inhibition of GluN2A-containing receptors (Appendix A). The observations that decay times and TCN-201 inhibition of GluN1/2A/2B^R540H^ and GluN1/2A/2B^R696H^ are similar to WT GluN1/2A/2B receptors corroborate the interpretation of data from Figure 2 and Figure 3, suggesting that GluN2B mutants reduce the contribution of GluN1/2B receptors to the NMDA-EPSC due to relatively poor synaptic incorporation of these mutant diheteromeric GluN1/2B receptors. However, the native GluN2A subunit rescues this deficit by enabling synaptic incorporation of the mutant GluN2B subunits into mutant GluN1/2A/2B receptors with decay time and TCN-201 inhibition comparable to WT GluN1/2A/2B receptors.

### 3.4. Different Consequences of the Same Missense LoF Mutation in GluN2A and GluN2B on the Time Course of NMDA-EPSCs

A striking finding from our GluN2B mutant molecular replacement experiments shown in Figure 2, and from our analogous GluN2A mutation results [40], is that the direction of effects of GoF and LoF mutations is the same. Nonetheless, GluN2B mutations seemed to have the opposite effect on NMDA-EPSCs to GluN2A mutations [40]. For example, mutant GluN2A NMDA-EPSCs had slower decays while mutant GluN2B NMDA-EPSCs had faster decays, regardless of the classification of the mutations as GoF or LoF.

To further evaluate the importance of the GluN2 subunit identity for the synaptic phenotype of the mutants, we studied the effect of the same mutation in GluN2A and GluN2B. We previously evaluated the synaptic effect of a disulphide-bridging cysteine mutant, C436R, in the subunit GluN2A. Here, we take advantage of the observation that some patients also have the same missense mutation, C436R, in GluN2B [42]. We found that the C436R mutation had strikingly different effects depending on in which GluN2 subunit it resided. While peak amplitudes of NMDA-EPSCs appeared similar for GluN2A^C436R^ and GluN2B^C436R^ (Figure 5bi, Appendix A: *F* (1,31) = 0.18, *p* < 0.67, BF_10_ = 0.368), they differed significantly in their decay kinetics (Figure 5bii, Appendix A: *F* (1,31) = 40.71, *p* < 0.001, BF_10_ = 5.47 × 10^6^) and charge transfer (Figure 5biii, Appendix A: *F* (1,31) = 19.73, *p* < 0.001, BF_10_ = 468), with about a four-fold difference (Figure 4). In summary, the data suggest that the identity of the mutated GluN2 subunit is an important determinant of the effects on synaptic NMDA-EPSCs that should be considered in addition to the classification of the mutation as GoF or LoF.

### 3.5. Partial Haploinsufficiency for a Grin2b Loss-of-Function Allele

The above experiments were performed by replacing GluN2B expressed from both alleles of the endogenous *Grin2b* gene with mutated human GluN2B. However, patients with rare *GRIN2B* mutations tend to be heterozygotes. What effect can we expect in heterozygotes for LoF alleles? How much can we expect GluN2A subunits to compensate for the loss of GluN2B? We addressed these questions by examining the effect of homozygous and heterozygous null alleles on evoked NMDA-EPSCs. We achieved this genetic manipulation by transfecting Cre-recombinase in pyramidal neurons in CA1 of organotypic slices prepared from *Grin2b*^+/+^, *Grin2b*^*fl*/+^ and *Grin2b^fl/fl^* mouse pups. We found evidence for differences among the genotypes with respect to NMDA-EPSC charge transfer (Figure 6ai,b; Appendix A, *F* (2,104) = 61.7, *p*  =  <.001, BF_10_  =  1.6 × 10^17^) and peak amplitude (Figure 6ai,c; Appendix A, *F* (2,104) = 17.57, *p*  =  <.001, BF_10_  =  2.04 × 10^5^), however, we could not find any difference between WT and heterozygotes in post hoc comparisons (charge: Figure 6b; Appendix A, *t*(104) = 0.41, *p*  = 0 .68; peak: Figure 6c; Appendix A, *t*(104) = −0.53, *p*  = 0 .60;). In contrast, while decay time constants also differed significantly among the genotypes (Figure 6aii,d; Appendix A, *F* (2,104) = 35.84, *p*  ≤ 0.001, BF_10_  =  9.43 × 10^11^), we found that the heterozygote NMDA-EPSC decay time course differed significantly from WT, with heterozygote and homozygote decay time constants at 81% (95% CIs [69–96%]) and 50% (95% CIs [42–59%]) that of WT levels (100%) (Figure 6d). Rise times of the NMDA-EPSCs differed significantly between genotypes (Figure 6e; Appendix A, *F* (2,104) = 6.50, *p*  =  0.002, BF_10_  =  28.2), with a significant linear downward trend (Appendix A, *F* (1,104) = 12.72, *p*  ≤ 0.001), although we could not resolve a significant difference specifically between heterozygotes and WT (Appendix A, *t*(104) = −1.45, *p*  =  0.15). The feature of NMDA-EPSCs most obviously affected in heterozygotes for a GluN2B null allele was an acceleration of the decay and thereby a shortening of the time course, indicating partial haploinsufficiency. Together, these data suggest that GluN2A can compensate for a decrease in, but not complete absence of, GluN2B expression.

## 4. Discussion

Our investigation reveals mechanisms by which some mutations from patients with *GRIN2B*-related disorders result in defective synaptic transmission. We demonstrated that synaptic delivery of GoF GluN2B mutants R696H and R540H is reduced compared to WT GluN2B in *Grin2a*^−/−^*b*^−/−^ neurons, whereas synaptic delivery of a LoF mutant, C456Y, is severely attenuated (Figure 1). Furthermore, GoF GluN2B mutants R696H and R540H prolong the decay of NMDA-EPSCs in *Grin2a*^−/−^*b*^−/−^ neurons (Figure 1). By contrast, all mutants were able to rescue NMDA-EPSC peak amplitudes compared to WT GluN2B in *Grin2b*^−/−^ neurons, suggesting normal synaptic delivery in the presence of native GluN2A (Figure 2). In addition, all GluN2B mutants tested accelerate NMDA-EPSC decays in *Grin2b*^−/−^ neurons, which to varying extents is accompanied by a reduction of NMDA-EPSC charge transfer (Figure 2). Evaluation of functional properties for diheteromeric and triheteromeric NMDA receptors expressed in HEK293T cells suggest that effects of GoF GluN2B mutations are virtually abated by coassembly with GluN2A into triheteromeric GluN1/2A/2B receptors (Figure 4). These results are consistent with a mechanism in which the GluN2B mutants primarily reduce the contribution of diheteromeric GluN1/2B to the NMDA-EPSC and shift the combination of synaptic NMDA receptors towards GluN1/2A and GluN1/2A/2B receptor subtypes (Figure 7). We then demonstrated that a disease-associated putative LoF missense mutation can produce opposing effects on NMDA-EPSCs depending on whether the mutated subunit is GluN2A or GluN2B (Figure 5). Finally, we found that heterozygosity for a GluN2B null allele can lead to effects on the kinetics of NMDA-EPSC properties that are intermediate between homozygotes for WT or null alleles (Figure 6).

The functional incorporation of GluN2B mutants in *Grin2a*^−/−^*b*^−/−^ CA1 neurons in Figure 1 is broadly consistent with previous investigations in heterologous expression systems [42] (Table 1). Swanger et al. reported that GluN1/2B^R540H^ and GluN1/2B^R696H^ receptors expressed in HEK cells showed a 2.4- and 1.3-fold lower surface expression than WT, coinciding with a 1.2- and 4.1-fold lower current density for GluN1/2B^R540H^ and GluN1/2B^R696H^ receptors, respectively [42]. However, charge transfer from GluN1/2B^R540H^ and GluN1/2B^R696H^ receptors in their experiments were both ~1.4-fold greater than WT due to the respective 2.0- and 3.6-fold slower current deactivation (Table 1). Here, we found a more modest slowing of current deactivation, 1.9- and 1.4-fold for GluN1/2B^R540H^ and GluN1/2B^R696H^ receptors, respectively (Figure 4; Appendix A). This apparent discrepancy in current deactivation between studies is likely a result of different experimental conditions. Swanger et al. [42] measured deactivation following long 1.5 s applications of glutamate, which can result in pronounced receptor desensitization, whereas our study used brief, synaptic-like 5 ms glutamate applications that minimize receptor desensitization. Consistent with our results from NMDA receptors expressed in HEK293T cells and 5 ms glutamate applications, measurements of NMDA-EPSCs mediated by GluN1/2B^R540H^ and GluN1/2B^R696H^ revealed smaller peak amplitudes and only modestly slower NMDA-EPSC decays compared to WT, coinciding with an overall charge transfer that was slightly lower than WT. Our assessment of functional incorporation of GluN1/2B^C456Y^ supports the observations by Swanger et al. [42] of low surface expression and small currents in heterologous expression systems for this mutant.

While the functional incorporation experiments in Figure 1 are useful in assessing the potential for mutant receptors to reach the synapse (as GluN1/2B diheteromers), it alone does not provide an indicator of the impact these mutations have on NMDA-EPSCs. Many of the synaptic NMDA receptors in juvenile and adult CA1 neurons are thought to be composed of triheteromeric GluN1/2A/2B receptors [61], and prediction of the effect of mutations on subunit assembly and function of triheteromeric receptors is not straightforward. One striking finding from our mutant rescue experiments was the difference in effect of GoF mutations on NMDA-EPSCs with (Figure 2) and without (Figure 1) native GluN2A. A key finding that contributes to the discrepancy is that the GoF effects of GluN2B R540H or R696H were obscured upon assembly with GluN2A into triheteromeric GluN1/2A/2B receptors (Figure 4). While the dominant effect of GluN2A has been demonstrated for the kinetics of WT triheteromeric receptors [24,58,59], this finding suggests that GluN2A dominance can overcome the GoF effects of GluN2B mutants located in the agonist binding domain. This is in contrast to triheteromers with mutations in the GluN2B ion channel leading to dominant or intermediate GoF effects by alleviating Mg^2+^ block and thereby enhancing NMDA-receptor-mediated currents at negative holding potentials [62].

While the conclusions in this study share similarities to our work on GluN2A mutants [40], there will be limits to the extent to which we can generalize the findings to other mutations. In both studies, here and in [40], we focused on missense mutations located within the agonist binding domain. The ability of agonist binding to control channel gating has been shown to be a quality control checkpoint for the endoplasmic reticulum (ER)-exit of GluN2B [42,63], as it is for other glutamate receptors [64,65,66,67,68,69,70]. This coupling between agonist-binding domain functionality and forward trafficking of the receptor may explain why surface expression of NMDA receptors is often perturbed by agonist binding domain mutations [42]. In contrast, mutations occurring in the ion channel portion of the NMDA receptor generally appear to have less of an effect on surface delivery of the receptor [71]. Indeed, pore mutations can also affect magnesium block and could have gain-of-function effects on NMDA-EPSCs that are very different to the effects we observed here for R696H and R540H [62,72,73].

Our investigation of GluN2B mutation effects involved expressing mutant *GRIN2B* cDNA in homozygote *Grin2b*^−/−^ neurons. Since patients with a *GRIN2B*-related disorder are heterozygous for a mutant allele, our experiments may overestimate the effect that mutated GluN2B could have on synaptic currents. We began to address this potential shortcoming by measuring the properties of NMDA-EPSCs in neurons with different genotypes of a null *Grin2b* allele. Most effects that we observed in *Grin2b* knockout neurons were strongly diminished in heterozygotes. Similarly to *Grin2a* heterozygotes [40], intermediate effects were most apparent for the NMDA-EPSC decay of neurons heterozygous for the null allele, but overall, the effects of heterozygosity on NMDA-EPSCs seemed modest.

So how could *GRIN2B* mutations have such profound consequences and lead to disease? Recently, mice heterozygous for either a null allele or one of the LoF missense mutations studied here, GluN2B-C456Y, were generated and characterized [74]. Consistent with our findings, the NMDA-EPSCs in CA1 neurons from these heterozygote mice had accelerated decay kinetics. While these mice showed enhanced anxiety-like behaviour as pups, their behaviour switched to more anxiolytic or hypoactive as the mice developed into juveniles and adults. This suggests that the impact of GluN2B mutations depends on the developmental stage. At a cellular level, heterozygosity for mutations in *GRIN2B* have also been shown to affect neuronal differentiation [75], and overexpression of GluN2B mutants has been shown to affect dendritic morphogenesis [76,77], both effects which we likely evade in our postnatal (< P7), single-cell genetic manipulation. Nonetheless, our data suggest that very different deficits in NMDA-EPSCs may be apparent depending on developmental stage. The subunit composition of synaptic NMDA receptors changes in most neuronal cell types during brain development [2]. In hippocampal CA1 pyramidal neurons, this developmental shift is accompanied by an acceleration of NMDA-EPSCs, since the upregulation of GluN2A expression produces triheteromeric GluN1/2A/2B receptors with faster decay times, while the contribution from GluN1/2B receptors with slower decay times is decreased [49,78,79,80]. Thus, the developmental transition from primarily diheteromeric GluN1/2B to also including triheteromeric GluN1/2A/2B receptors may be reminiscent of NMDA-EPSCs shown in Figure 1, which lack native GluN2A, and NMDA-EPSCs in Figure 2 that include native GluN2A subunits. Hence, the effects of some GluN2B mutations in CA1 principal neurons may be more pronounced and may differ between LoF and GoF GluN2B mutations early in development before the expression of GluN2A can mitigate functional deficits in triheteromeric GluN1/2A/2B receptors.

Triheteromeric receptors can also be formed from other combinations of GluN2 subunits, where the impact of GluN2B mutations could be different from those observed here (i.e., GluN1/2A/2B receptors in CA1 pyramidal neurons). While the deactivation kinetics of GluN1/2A/2B receptors are dominated by GluN2A, GluN2B has a greater influence on deactivation kinetics when co-assembled with GluN2D into GluN1/2B/2D triheteromers [25]. GluN2B mutants may therefore have a greater impact on the function of GluN1/2B/2D triheteromers, especially since at rest, GluN1/2B/2D receptors are less attenuated by ambient extracellular Mg^2+^ [25]. If this is the case, then mutant GluN1/2B/2D receptors may be more pathogenic in affecting the excitability of a variety of other cell populations including hippocampal interneurons [81,82,83] and granule cells [84], striatal medium spiny neurons [85], cerebellar Golgi cells [86], and neurons within the subthalamic nucleus [87], substantia nigra pars compacta [88,89,90,91] and spinal cord [92].

In summary, our results demonstrate that functional incorporation of mutant GluN2 subunits into synaptic receptors, the presence of triheteromeric NMDA receptors, and the identity of the mutated GluN2 subunit can influence the functional classification of NMDA receptor variants as GoF or LoF mutations. These findings highlight the complexity of interpreting effects of disease-causing NMDA receptor missense mutations in the context of neuronal function, and provide important molecular and cellular insights into the mechanisms by which disease-causing GluN2B mutations can affect synaptic transmission.

## Figures and Tables

**Figure 1 brainsci-12-00789-f001:**
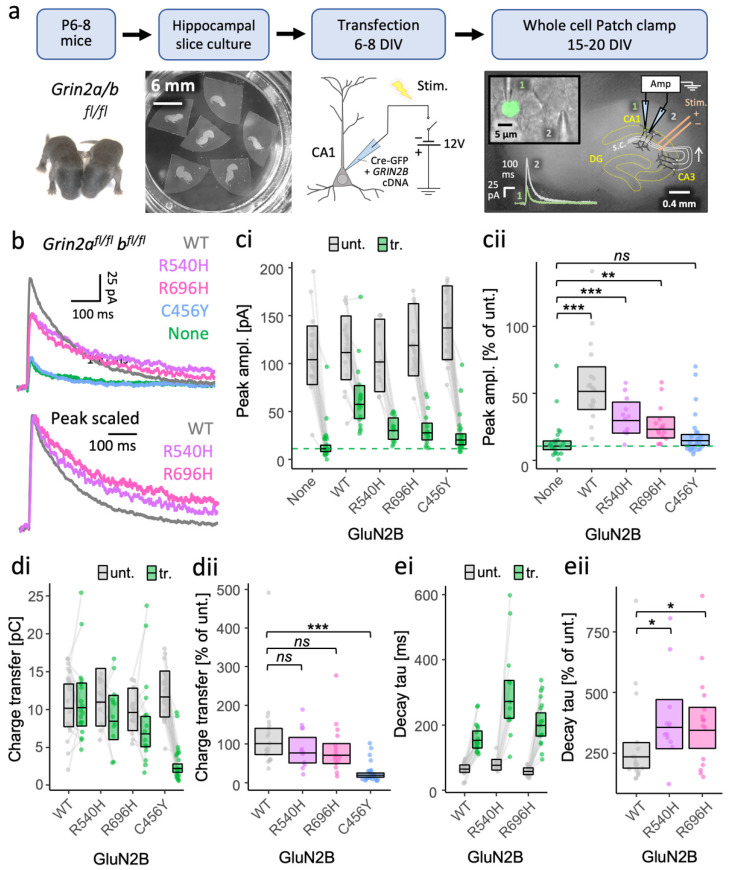
More effective rescue of NMDA-EPSCs in GluN2A/B double knockout neurons by putative GoF mutants (R696H and R540H) than by LoF mutation (C456Y). (**a**) Experimental protocol used to test functional incorporation of GluN2B mutants. Plasmids expressing Cre-GFP and human *GRIN2B* cDNA were cotransfected into CA1 pyramidal neurons of organotypic hippocampal slices from *Grin2a^fl^/^fl^**b^fl/fl^* mice. With transfection, the action of Cre-GFP at floxed *Grin2a* and *Grin2b* alleles knocks out native mouse GluN2A and 2B protein expression concurrently with expressing human GluN2B variants (WT or mutant). Untransfected neurons continue to express native NMDA receptors (e.g., GluN2A and GluN2B). (**b**) Representative NMDA-EPSC_+20 mV_ (average of 30 sweeps) from transfected CA1 neurons. (**c**,**d**) NMDA-EPSC_+20 mV_ peak amplitudes and charge transfer, respectively, for *Grin2a^fl/fl^b^fl/fl^* (untransfected, *abbrv.* Unt.) neurons, and *Grin2a*^−/−^*b*^−/−^ neurons rescued with human GluN2B WT, GoF (R540H, R696H) or LoF (C456Y) mutants (transfected, *abbrv.* Tr.). (**e**) NMDA-EPSC_+20 mV_ decay time constant (tau) and charge transfer, respectively, for *Grin2a^fl/fl^b^fl/fl^* neurons, and *Grin2a*^−/−^*b*^−/−^ neurons rescued with human GluN2B WT, and GoF mutants R540H and R696H; (**c**–**e**) (i) data points of measurements made in individual neurons; matched data points, for simultaneously recorded untransfected (unt.) and transfected (tr.) neurons, are connected by a line; (ii) response ratios (transfected/untransfected) are expressed as a percentage and plotted for each pair of transfected-untransfected neurons. Crossbars in (i) and (ii) show the estimated marginal means with 95% confidence intervals backtransformed from the linear mixed models (Appendix A). Post hoc pairwise comparisons were made using Dunnett’s stepdown procedure. Standardised effect sizes (*r*) for comparisons of response ratios for: (**cii**) peak amplitudes (vs control: no *GRIN2B* cDNA, *a.k.a.* None) were +0.62, +0.39, +0.32 and +0.10 for WT, R540H, R696H, and C456Y, respectively (N = 90); or (**dii**) charge transfer (vs control: WT) was −0.12, −0.17 and −0.68 for R540H, R696H, and C456Y, respectively (N = 71) of (**eii**); decay time constant (tau) was +0.33 and +0.33 for mutants R540H and R696H, respectively (N = 47). Data for the NMDA-EPSC amplitudes in *Grin2a*^−/−^*b*^−/−^ neurons with and without WT GluN2B were published in [40]. *ns* = not significant (at α = 0.05), * = *p* < 0.05, ** = *p* < 0.01, *** = *p* < 0.001.

**Figure 4 brainsci-12-00789-f004:**
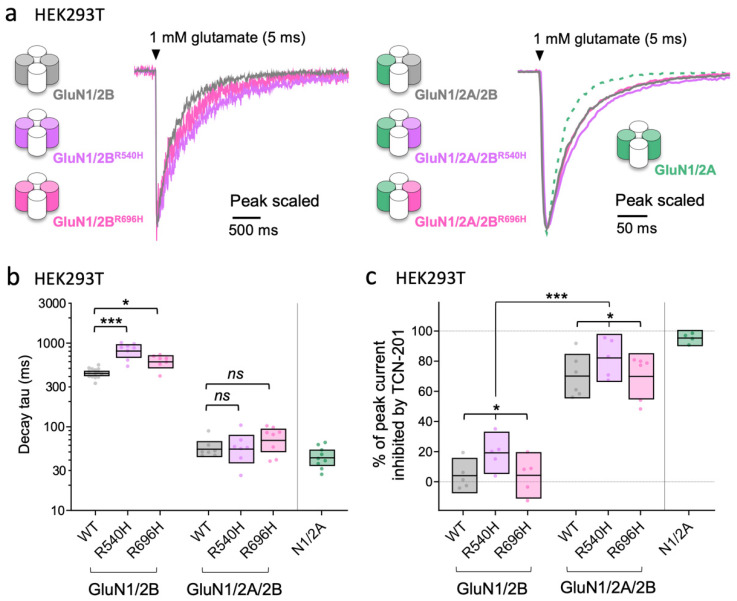
Assembly of mutant GluN2B with GluN2A in triheteromeric GluN1/2A/2B receptors can abate functional effects of GoF missense GluN2B variants. (**a**) Representative fast-application whole-cell patch-clamp recordings of NMDA receptor responses normalized to the peak amplitude. The NMDA receptor subtypes are expressed in HEK293T cells and responses are activated by brief 5 ms exposures to 1 mM glutamate in the continuous presence of 100 μM glycine. (**b**) Decay time constants for WT and mutant NMDA receptor subtypes. Crossbars represent the geometric mean and 95% confidence intervals. The statistical comparisons presented are Dunnett’s post hoc test results following a significant mutation x subtype interaction (*F* (2,43) = 6.66, *p* = 0.003) from a two-way ANOVA (Type III) on log_10_-transformed decay tau values. (**c**) Inhibition by 5 µM TCN-201 of response amplitudes from WT and mutant NMDA receptor subtypes activated brief 5 ms application of 1 mM glutamate in the continuous presence of 3 μM glycine. Crossbars represent the mean and 95% confidence intervals. The statistical results presented in (**c**) are for the main effects of mutation (*F* (2,26) = 3.91, *p* = 0.033) and subtype (*F* (1,26) = 206.6, *p* ≤ 0.001) from a two-way ANOVA (Type III) on the % of peak current inhibited by TCN-201; the mutation x subtype interaction was not statistically significant (*F* (2,26) = 0.045, *p* = 0.96). In both (**b**) and (**c**), factor levels for mutation were WT, R540H and R696H; factor levels for subtype were diheteromer (GluN1/2B) and triheteromer (GluN1/2A/2B). *ns* = not significant (at α = 0.05), * = *p* < 0.05, *** = *p* < 0.001.

**Figure 5 brainsci-12-00789-f005:**
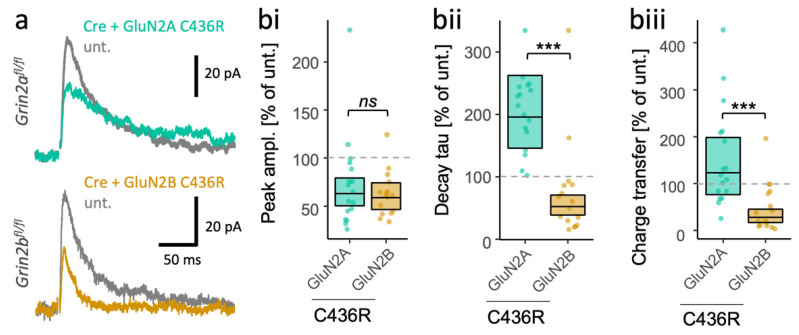
Subunit-dependent outcome of identical disease-associated LoF mutations in *GRIN2A* and *GRIN2B*. (**a**) Representative NMDA-EPSCs (average of 10 sweeps) from *Grin2b^fl/fl^* (untransfected abbrv. unt.) neurons, and *Grin2b*^−/−^ neurons rescued with human GluN2A^C436R^ or GluN2B^C436R^. (**b**) Ratios (transfected/untransfected) of the response (peak amplitude, decay time constant or charge transfer) are expressed as a percentage and plotted for each pair of transfected-untransfected neurons. Crossbars in (ii) and (iii) show the estimated marginal means with 95% confidence intervals backtransformed from the linear mixed models (LMM). The *p*-values from the hypothesis tests are derived from the same LMM (Appendix A). Standardised effect sizes (*r*) for GluN2 subunit background (of the C436R mutation) on response ratios were **(bi)** for peak amplitudes, −0.08; (**bii**) for decay taus, −0.75; and (**biii**) for charge transfer, −0.62 (N = 33). *ns* = not significant (at α = 0.05), *** = *p* < 0.001.

**Figure 6 brainsci-12-00789-f006:**
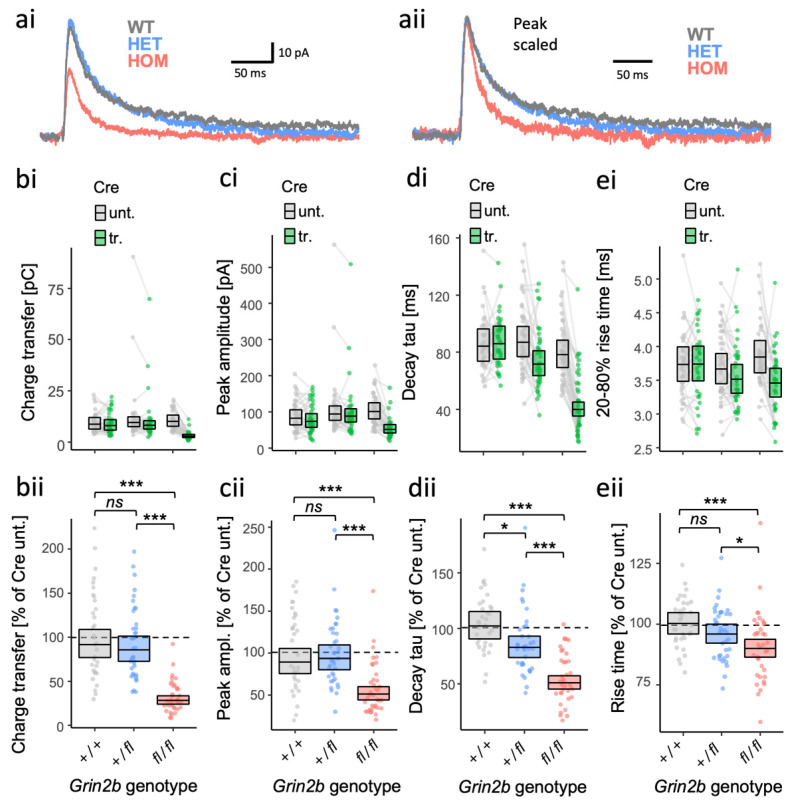
Dose-dependent effects of *Grin2b* null alleles on NMDA-EPSC kinetics. (**a**) Representative NMDA-EPSCs from *Grin2b*^+/+^, *Grin2b*^+/−^ or *Grin2b*^−/−^ neurons. NMDA-EPSC_+20 mV_ charge transfer (**b**), peak amplitude (**c**), decay time constant (**d**) and 20–80% rise-time (**e**) in *Grin2b*^+/+^, *Grin2b*^+/*fl*^ or *Grin2b^fl/fl^* (untransfected) neurons and *Grin2b*^+/+^, *Grin2b*^+/−^ or *Grin2b*^−/−^ neurons (transfected, with Cre-GFP). (**b**–**e)** (i) Data points of measurements made in individual neurons. Matched data points, for simultaneously recorded untransfected and transfected neurons are connected by a line; (ii) response ratios (transfected/untransfected) are expressed as a percentage and plotted for each pair of transfected-untransfected neurons. Crossbars in (i) and (ii) show the estimated marginal means with 95% confidence intervals backtransformed from the fitted linear mixed models (Appendix A). Post hoc pairwise comparisons were made according to the Westfall stepwise procedure. Standardised effect sizes (*r*) for comparisons of each genotype with WT for response ratios, for heterozygous and homozygous genotypes, respectively, were (**aii**) for charge transfer, −0.05 and −0.69; (**bii**) for peak amplitude, +0.04 and −0.43; (**cii**) for decay taus, −0.24 and −0.63; and (**dii**) for rise-time, −0.14 and −0.33, (N = 107). *ns* = not significant (at α = 0.05), * = *p* < 0.05, *** = *p* < 0.001.

**Figure 7 brainsci-12-00789-f007:**
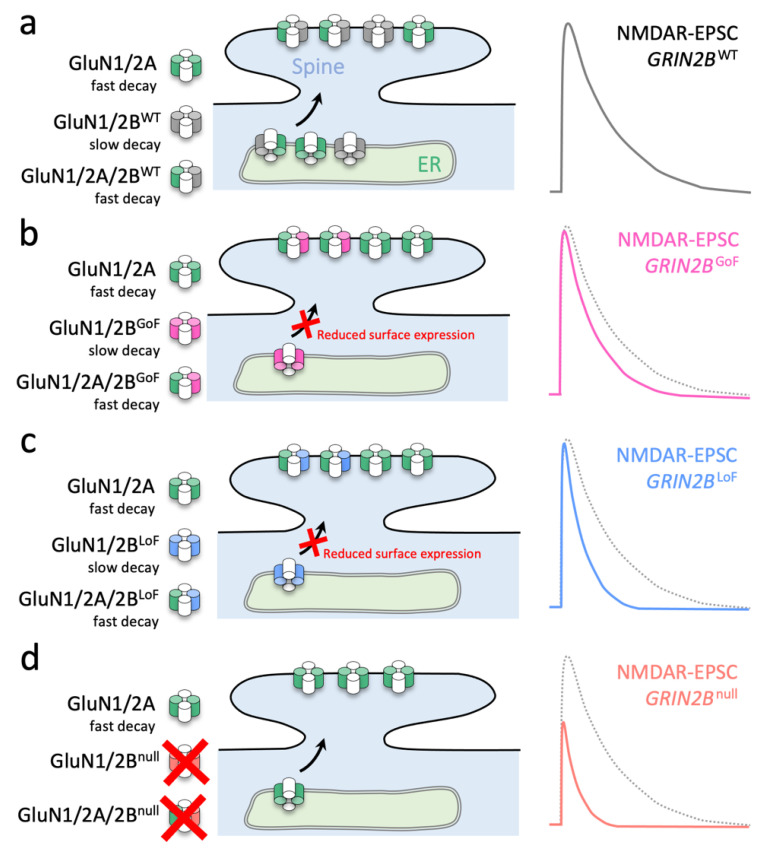
Scheme summarizing how different molecular defects in the agonist binding domain of GluN2B converge to accelerate NMDA-EPSCs in CA1 neurons. (**a**) Schaffer collateral synapses onto CA1 neurons contain triheteromeric receptors (GluN1/2A/2B) and a small population of diheteromeric receptors (GluN1/2A and GluN1/2B). (**b**) GoF missense mutations in GluN2B cause GluN1/2B receptors to have more prolonged decay (Figure 1b,e), but do not traffic effectively to synapses (Figure 1b,c). Synaptic GluN1/2A/2B receptors with GoF GluN2B mutations traffic comparatively better than their GluN1/2B counterparts (Figure 2a), but the NMDA-EPSC time course is accelerated owing to the dominance of GluN2A on the deactivation of triheteromeric NMDA receptors (Figure 4b). (**c**) LoF missense mutations (that strongly reduce GluN2B surface expression) lead to an absence of mutant GluN1/2B receptors (Figure 1b,c), and likely also a greater representation of GluN1/2A receptors at synapses (Figure 2a). (**d**) Genetic deletion (i.e., the null allele) of *Grin2b* prevents the formation of NMDA receptors with any GluN2B subunits. GluN2A cannot fully compensate for the loss of (both) *Grin2b* alleles (Figure 6b). Since synapses in neurons with either GoF or LoF GluN2B missense mutations have fewer GluN1/2B diheteromers, they both exhibit more extensive inhibition of their associated NMDA-EPSCs by TCN-201 (compared to WT) (Figure 3).

**Table 1 brainsci-12-00789-t001:** Disease-associated mutations in the agonist-binding domain of GluN2 subunits investigated in this study. Functional data represent fold changes in receptor activity by gene variant. For example, a value of −4.1 is a 4.1-fold decrease in the respective measure of variant receptor activity compared to WT (i.e., no mutation). A value of 1.0 means no change from WT whereas n.d. indicates that the measurement was not made because sufficient currents could not be detected (e.g., due to the surface expression of the mutant being too low). See the footer notes for the sources of information presented in this table. None of the mutants (that produce sufficiently large currents) have been shown to have any discernible effect on Mg^2+^ block [45]. * Data relating to phenotypes of *GRIN2B* and *GRIN2A* mutations were obtained from [34,46], respectively. ^$^ Fold changes for surface expression, current density, deactivation, Glu/Gly potency and charge transfer were calculated from published data obtained by expression of mutant GluN1/2A or GluN1/2B in heterologous expression systems (either oocytes or HEK cells) [42].

Subunit	Mutation	Phenotype *	Surface Expression ^$^	Current Density ^$^	Deactivation ^$^	Glu Potency ^$^	Gly Potency ^$^	Charge Transfer ^$^	Class
** *GRIN2B* **	R540H	ID, fSz (ASD, gSz)	−2.4	−1.2	+2.0	2.3	+1.5	+1.4	**GoF**
	R696H	ID, ASD (gSz)	−1.3	−4.1	+3.6	4.5	−1.2	+1.5	**GoF**
	C456Y	ID, ASD	−11.8	−1367	n.d.	3.8	−2.6	n.d.	**LoF**
	C461F	ID, ASD, gSz	−6.1	−9.8	−20.4	−113	2.5	−171	**LoF**
	C436R	ID, ASD, fSz, gSz	−8.7	−820	n.d.	n.d.	n.d.	n.d.	**LoF**
** *GRIN2A* **	C436R	gSz	−66.7	−3400	n.d.	−1.5	+1.5	n.d.	**LoF**

Abbreviations: n.d.: See definition above in the Table 1 legend. ID: Intellectual disability, fSz: focal seizures, ASD: Autism Spectrum Disorder, gSz: generalized seizures, GoF: gain-of-function, LoF: loss-of-function, Glu: L-glutamate, Gly: Glycine.

## Data Availability

Data and code used for analysis are deposited at https://github.com/acp29/Elmasri_GRIN2B (accessed on 5 May 2022).

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
