# Peer review of "Synaptic Dysfunction by Mutations in GRIN2B: Influence of Triheteromeric NMDA Receptors on Gain-of-Function and Loss-of-Function Mutant Classification"

_brainsci, 2022, doi:10.3390/brainsci12060789_

Round 1
Reviewer 1 Report
In the present study, the authors revealed how some mutations from patients with GRIN2B-related disorders result in defective synaptic transmission. Using molecular replacement experiments in cultured hippocampal slices and functional properties for diheteromeric and triheteromeric NMDA receptors expressed in HEK293T cells, the authors demonstrated effects of GoF and LoF mutations.
The results presented are convincing and I only have a few minor comments.
1) Since mutations can affect NMDA receptor localization, data on whether AMPA/NMDA ratios changed would be informative. It follows from the methodology that AMPA currents were recorded on a regular basis.
2) Please add a chart about tau changes in Figure 3. Please verify if the time scale is correct in Fig. 3А.
3) A scheme summarizing how different molecular defects can converge on similar NMDAR-mediated EPSCs would be helpful in the discussion
4) A small commentary on the possible functional significance of the identified mutation properties for the pathogenesis of certain diseases would be interesting.
Author Response
Thank you to the reviewer for their comments and helpful suggestions. Please find below our point-by-point response.
1) Since mutations can affect NMDA receptor localization, data on whether AMPA/NMDA ratios changed would be informative. It follows from the methodology that AMPA currents were recorded on a regular basis.
We thank the reviewer for their suggestion and have now computed AMPA-NMDA ratios and included a phrase in lines 424-425 that summarises the statistics for the comparison of AMPA-NMDA ratios between the GluN2B variants (WT and mutants).
2) Please add a chart about tau changes in Figure 3. Please verify if the time scale is correct in Fig. 3А.
We thank the reviewer for this excellent suggestion. We have now measured the decay time constants for the NMDA-EPSCs before and after TCN-201. The graph of the data is shown in a new figure in the supplement, Fig. S4. The finding is insightful, and so we have added a paragraph relating to it in the results section, lines 520-536.
In answer to the last comment here from the reviewer, the time scale in Fig. 3A is correct – the examples chosen are representative of the group data in terms of the amount of block by TCN-201, but these examples all happen to have faster decays than the examples shown in Figures 1 and 2. In any case, to be more consistent with the formatting in some of the other figures, we have lengthened the scale bar and increased the time proportionally.
3) A scheme summarizing how different molecular defects can converge on similar NMDAR-mediated EPSCs would be helpful in the discussion
This is an excellent idea. We have added an additional (final) main text figure, Fig. 7, illustrating how the molecular defects could converge on similar NMDA-EPSCs. We have explained the model in the accompanying figure legend with reference to supporting evidence in other figures.
4) A small commentary on the possible functional significance of the identified mutation properties for the pathogenesis of certain diseases would be interesting.
We have added further discussion relating to the pathogenesis of neurodevelopmental disorders arising from GRIN2B mutations in lines 765-771 of the manuscript.
Reviewer 2 Report
Marwa Elmasri and colleagues report that several GoF and LoF mutations in GRIN2B influence NMDA functions. Particularly, GluN2B GoF and LoF mutants show similar phenotypes in the presence of GluN1/2A/2B triheteromers, while the LoF mutation, C456Y, shows much less rescue effect than GoF mutation in the absence of both GluN2A and GluN2B. The authors conclude that the presence of triheteromeric NMDA receptors, and the types of the mutated GluN2 subunit are all important factors that can influence the function of the NMDA variants. There are some concerns need to be addressed.
Even though fig.2aiii shows a comparable relative peak amplitude, which indicates both GOF and LOF rescue NMDA-EPSC in Grin2b-/- neurons, it looks to me that the absolute amplitude in fig.2aii shows a quite big difference in un-transferred groups which are supposed to be comparable, not as consistent as shown in fig.1ci. How to explain this discrepancy?
The supplementary figures are inaccessible.
The abstract is too long, which does not actually abstract the whole content very well. There are also a number of sentences that are so complicated for easy read. This part should be substantially polished into a shortened and precise style.
Author Response
Thank you to the reviewer for their comments and helpful suggestions. Please find below our point-by-point response.
1) Even though fig.2aiii shows a comparable relative peak amplitude, which indicates both GOF and LOF rescue NMDA-EPSC in Grin2b-/- neurons, it looks to me that the absolute amplitude in fig.2aii shows a quite big difference in un-transferred groups which are supposed to be comparable, not as consistent as shown in fig.1ci. How to explain this discrepancy?
It is hard to know exactly why the measurements of the internal control, untransfected neuron appear to vary between the experimental groups. While an effort was made to try and evoke synaptic responses with a consistent stimulus voltage setting, the effectiveness of the stimulus for evoking an NMDA-EPSC depends on, among other things, the condition of the stimulating electrode (e.g. resistance, stray capacitance, bubbles etc). It was rarely practical to record all mutant conditions in the same experiment, so variation in the stimulating electrode condition over the course of this series of experiments could potentially lead to some apparent differences between untransfected neurons with respect to their mean NMDA-EPSC peak amplitude and charge transfer. What this does serve to illustrate though is how important it was for us to use untransfected neurons as an internal control.
2) The supplementary figures are inaccessible.
The supplementary figures were within the same PDF so we are not sure why the author had trouble gaining accessing to the figures. Assuming the reviewer had trouble understanding the contents of the supplement, we have tried to simplify somewhat the text in the supplementary figure legends.
3) The abstract is too long, which does not actually abstract the whole content very well. There are also a number of sentences that are so complicated for easy read. This part should be substantially polished into a shortened and precise style.
We thank the reviewer for this comment. Indeed, there were many long sentences and the abstract did not capture all of our findings. We have rewritten the abstract to accommodate the reviewer’s suggestions. The abstract is now 311 words long (almost 40 words shorter than the original abstract), has shorter sentences, and includes some summary of the findings reported in all the main figures. The abstract now reads as follows:
GRIN2B mutations are rare but often associated with patients having severe neurodevelopmental disorders, including a varying range of symptoms such as intellectual disability, developmental delay and epilepsy. Patient symptoms likely arise from mutations disturbing the role that the encoded NMDA receptor subunit, GluN2B, plays at neuronal connections in the developing nervous system. In this study, we have investigated the cell-autonomous effects of putative gain- (GoF) and loss-of-function (LoF) missense GRIN2B mutations on excitatory synapses onto CA1 pyramidal neurons in organotypic hippocampal slices. In the absence of both native GluN2A and GluN2B subunits, functional incorporation into synaptic NMDA receptors was attenuated for GoF mutants, or almost eliminated for LoF GluN2B mutants. NMDA receptor-mediated excitatory postsynaptic currents (NMDA-EPSCs) from synaptic GoF GluN1/2B receptors had prolonged decays consistent with their functional classification. Nonetheless, in the presence of native GluN2A, molecular replacement of native GluN2B with GoF and LoF GluN2B mutants all led to similar functional incorporation into synaptic receptors, more rapidly decaying NMDA-EPSCs and greater inhibition by TCN-201, a selective antagonist of GluN2A-containing NMDA receptors. Mechanistic insight was gained from experiments in HEK293T cells, which revealed that GluN2B GoF mutants slowed deactivation in diheteromeric GluN1/2B, but not triheteromeric GluN1/2A/2B receptors. We also show that a disease-associated missense mutation, which severely affects surface expression, causes opposing effects on NMDA-EPSC decay and charge transfer when introduced into GluN2A or GluN2B. Finally, we show that having a single null Grin2b allele has only a modest effect on NMDA-EPSC decay kinetics. Our results demonstrate that functional incorporation of GoF and LoF GluN2B mutants into synaptic receptors and the effects on EPSC decay times are highly dependent on the presence of triheteromeric GluN1/2A/2B NMDA receptors, thereby influencing the functional classification of NMDA receptor variants as GoF or LoF mutations. These findings highlight the complexity of interpreting effects of disease-causing NMDA receptor missense mutations in the context of neuronal function.